# Motion Detection Using Tactile Sensors Based on Pressure-Sensitive Transistor Arrays

**DOI:** 10.3390/s20133624

**Published:** 2020-06-28

**Authors:** Jiuk Jang, Yoon Sun Jun, Hunkyu Seo, Moohyun Kim, Jang-Ung Park

**Affiliations:** 1Nano Science Technology Institute, Department of Materials Science and Engineering, Yonsei University, Seoul 03722, Korea; jiukjang@wearablelab.net (J.J.); yoonsun17@wearablelab.net (Y.S.J.); hunkyu1359@wearablelab.net (H.S.); moohyunkim@wearablelab.net (M.K.); 2Center for Nanomedicine, Institute for Basic Science (IBS), Seoul 03722, Korea; 3Graduate Program of Nano Biomedical Engineering (NanoBME), Advanced Science Institute, Yonsei University, Seoul 03722, Korea

**Keywords:** transistor, tactile sensor, pressure sensor, gesture recognition

## Abstract

In recent years, to develop more spontaneous and instant interfaces between a system and users, technology has evolved toward designing efficient and simple gesture recognition (GR) techniques. As a tool for acquiring human motion, a tactile sensor system, which converts the human touch signal into a single datum and executes a command by translating a bundle of data into a text language or triggering a preset sequence as a haptic motion, has been developed. The tactile sensor aims to collect comprehensive data on various motions, from the touch of a fingertip to large body movements. The sensor devices have different characteristics that are important for target applications. Furthermore, devices can be fabricated using various principles, and include piezoelectric, capacitive, piezoresistive, and field-effect transistor types, depending on the parameters to be achieved. Here, we introduce tactile sensors consisting of field-effect transistors (FETs). GR requires a process involving the acquisition of a large amount of data in an array rather than a single sensor, suggesting the importance of fabricating a tactile sensor as an array. In this case, an FET-type pressure sensor can exploit the advantages of active-matrix sensor arrays that allow high-array uniformity, high spatial contrast, and facile integration with electrical circuitry. We envision that tactile sensors based on FETs will be beneficial for GR as well as future applications, and these sensors will provide substantial opportunities for next-generation motion sensing systems.

## 1. Introduction

Since computer systems were first developed, they have been able to perform simple computing operations, as well as analysis and evaluation of the collected data. Gesture recognition (GR) is referred to as computer recognition and the digitization of human movement, such as the touch of a finger or the motion of a hand [1]. In recent years, to develop more spontaneous and instant interfaces between this system and users, technology has evolved that can design efficient and simple GR techniques. This advanced form of GR technology is essential for quick command systems and feedback between computers and people in real situations, such as computer games and machine control. As the computational speed and amount of data that can be processed increase, these interfaces can be applied to advanced functions, including big data-based behavior models and robotics, followed by expansion of GR applications to the public.

In terms of the data collection method, GR can be largely divided into contact and non-contact interfaces. In the case of non-contact interfaces, most are based on tracking visual signals [2,3]. Signal processing is conducted by recognizing a specific part of the user and consequently analyzing the pose of the object. However, this method requires extensive use of the camera roll (e.g., the photodetector), and it depends significantly on environmental changes, such as brightness and humidity. In addition, it provides poor signal accuracy and currently has many limitations for handling numerous signals in series. Therefore, this form of GR is used mainly with arranged gestures at a predetermined location. In contrast, the command system, with its contact interface, consists of direct contact between the user and machines. The invention of the keyboard and mouse as input devices suggests that the principal and standard means of human interaction with a machine is the detection of tactile stimulations. Generally, the system converts the signal created by human touch into a single datum, and executes the command by translating a bundle of data into a text language or by triggering a preset sequence as a haptic motion. Compared to vision-based signal processing, the touch-based contact interface has a similar processing flow, but has less complexity, a small signal, and substantially better intuition.

After the development of principal input devices, predominantly keyboards, user interface technology has been developed and refined further with the emergence of a smartphone implemented with a touch sensor that enables the provision of information for the machine (i.e., the circuitry system) by a touch of a finger [4,5]. Among future technologies, GR has attracted attention as a key technology for facile communications with machines. Due to these industrial and research trends, research has been accelerated on motion sensing systems, which consist of tactile detection in various fields, such as robotics [6,7], artificial intelligence (AI) [8,9], human-machine interface (HMI), [10,11,12], and Internet of Things (IoT) [13]. Users can perform complex tasks through the provision of motion, including simple touch, and such systems can also exhibit very high efficiency, particularly in terms of augmented reality (AR) and virtual reality (VR). The characteristics required for effective motion sensing are (1) the sensitivity of the motion sensing device, (2) the size of the threshold for tactile stimulus, and (3) the speed with which the device can respond to the stimulus. These characteristics are considered important for fundamental GR performance. For example, the device developed for a GR application must be able to detect several tens of kilopascals (kPas) because the tactile pressure of a gentle human touch is in the range of 10 to 100 kPa. In addition, the realization of 3D touch technology, which can convert the magnitude of tactile pressures to other factors, such as the thickness of a line, should consider the sensitivity factor that affects how many steps the pen pressure can be divided into. Numerous studies have been reported on the assumption that the target application is based on a pen or finger touch, and have focused mainly on measuring more sensitive and wider ranges. In addition, the time delay from the moment of tactile touch until the device responds and the reliability of multiple contacts are significant factors in assessing the validity of the sensor.

Recently, tactile sensors for GR have been developed to provide convenience to users and acquire additional information by integrating various functions in addition to the detection of touch. As an example, a device exists that can detect a normal force and a shear force simultaneously, thereby providing detailed information about human touch. In addition, integrated tactile sensors for visual and auditory feedback to users are attracting attention in the field of human–machine interfaces [14]. Proximity sensors that can detect approaching objects can also be representative tactile sensors for sensing motion. Their principle follows that of a contact interface-based tactile sensor, but can also be used as a non-contact interface. In addition, advanced fabrication techniques have been studied in order to produce a highly-integrated, multi-functional sensor device [15,16,17,18]. As the development of flexible and stretchable electronics-based technologies has accelerated, tactile sensors are being developed that can operate even in harsh environments where reliable mechanical deformations, such as bending and stretching, are required [19,20,21,22,23]. These characteristics will provide a substantial technical basis for next-stage augmented reality and virtual reality (Figure 1).

## 2. Tactile Sensor Arrays

Tactile sensors can collect comprehensive data on various motions of the human body, from the touch of a fingertip to the movements of the entire body. The sensor devices have different characteristics that are important for target applications, and can be fabricated using various principles. The forms include piezoelectric [26,27], capacitive [28,29,30], piezoresistive [31,32] and field-effect transistor types, depending on the parameters to be achieved [33,34,35,36]. Here, we present basic information about the tactile sensor array device, which is used extensively as a tool for recognizing gestures, so that readers can gain a better understanding of the fundamental mechanisms involved.

### 2.1. Parameters of Tactile Sensors and Their Arrays

A tactile sensor is a device that acquires the applied force quantitatively as an electrical signal or some other indicator. Parameters for evaluating tactile sensors include their sensitivity, range of detectable pressure, minimum sensing pressure, response and recovery times, and reliability [33]. Sensitivity is the most intuitive parameter that can display the accuracy and characteristics of the tactile pressure sensor. The sensitivity of the pressure sensor is defined as follows:*S* = Δ*X*/Δ*P*(1)
where *S* is sensitivity, *X* is an arbitrary conversion signal (e.g., current, voltage, or capacitance), and *P* is pressure. High sensitivity suggests that a high-magnitude conversion signal can be obtained with only a small change in the input signal, which is an advantage when fine pressures must be measured. From the perspective of the recognition of motion and gestures, human motion can include a comprehensive range of movements, ranging from a feathery touch of the fingertip to large movements, such as footsteps or jumping. Therefore, it is better for the detectable tactile pressure range for data acquisition to cover a wide range, i.e., from subtle pressure to high pressure for the target application, because this will provide advantages in the use of a pressure sensor for sensing motion.

Response time is considered to be an important factor for real-time and repetitive detection, and refers to the time that is required to reach more than 90% of the output signal after the application of tactile pressure. Generally, pressure sensors can be approximated by first-order differential equations for time as follows:(2)$$X=X_{max}\left(1-e^{-\frac{t}{c}}\right)$$
where *X* is an arbitrary conversion signal, *X_max_* is 100% of the conversion signal for a specific stimulus, *t* is time, and *c* is a time constant for the sensor device.

For the recognition of motion, a tactile sensor device should be integrated as a large-scale or high-resolution two-dimensional (2D) array. The array-type pressure sensor scans a plurality of pixels over time, and this array device, which has a large number of pixels, may have limitations due to the time delay associated with the successive scan times for each of the pixels. Therefore, fast response time and the fast recovery time of individual pixels are essential to acquire array data simultaneously. In the case of an array device, the uniformity of the previously-mentioned parameters (e.g., sensitivity and range of detectable pressures) should be considered when the device is used in industrial applications.

As advanced, human-friendly electronics evolve, including wearable electronics, robotics, and artificial intelligence, motion sensing devices are being integrated radially into wearable devices with flexible or skin-attachable forms, and this can require a high level of reliability against mechanical deformation and cyclic folding and stretching.

### 2.2. Various Types of Tactile Sensors

Tactile sensors for converting tactile stimuli into electrical signals (e.g., current, voltage, capacitance) are being developed based on various principles, such as piezoresistive, piezoelectric, capacitive, optical, and transistors, in accordance with the purpose of the target application. Figure 2 shows the fundamental schematics of the types of tactile sensors with their transduction principles.

The transduction principle of a piezoresistive tactile sensor is to convert the change in the resistance of the device due to the application of external pressure into an electrical signal, with the current flowing mainly along the sensors. This mechanism is similar to that of a resistive tactile sensor, which utilizes change in contact resistance between two conductors upon physical force. These devices are fabricated from simple interlocked microstructures where physical deformation leads to change in contact resistance. Devices utilizing the piezoresistive principle have been studied in various forms and materials due to their excellent linearity, simple structure, method of operation, and the convenience of processing the output signal due to the facile mechanism [37,38,39,40]. Piezoresistive tactile sensors are suitable for measuring large pressure stimuli. These devices are mainly made using the complicated serpentine pattern of thin-film metal or the changes in the conducting path by three-dimensional percolation networks of one-dimensional or two-dimensional metal filler, such as metal nanoparticles and nanowires, within the elastomer matrix, inducing the resistance changes that are followed by current changes. Generally, compared to other types of devices, piezoresistive tactile sensors have fast response time and wide ranges in which they can detect pressure.

Piezoelectric-type tactile sensors use materials that induce changes in the internal electric dipole moment due to the pressure as their active materials [41]. This type of sensor has relatively high sensitivity and fast response time, and such sensors easily can be utilized for measuring dynamic pressure, such as vibration and friction [42,43,44]. A tactile sensor using a triboelectric material also generates triboelectric potential due to the friction and electrification with electrostatic induction, which broadly can be included in the piezoelectric method [45,46]. A triboelectric material-based device can detect various types of forces, including normal and shear forces, because of the friction-based potential changes within the material. However, these active materials are generally affected by external environments, such as temperature, because, in most cases, their piezoelectric properties are accompanied by pyroelectric properties [47], so these limitations must be solved when the device is used as a wearable device.

Optical operation mechanisms, such as light wavelength, phase, polarization, color, and intensity, are other mechanisms for flexible pressure sensors [48,49,50]. A tactile sensor using an optical mechanism prevents stray capacitances and thermal noise, and has lower crosstalk and a high spatial solution. Despite of these properties, medium power consumption is a representative problem of optical mechanisms, so this problem should be addressed for the future technology of tactile sensors [51].

Capacitive devices detect the tensile stimuli by sensing the change in capacitance that naturally is formed in an insulating layer between two conductive plates [52,53]. When the thickness of a material changes due to the pressure that is applied between two parallel plate electrodes, the distance between the two plates is changed by physical contact, and this results in a change in capacitance due to changes in the distance between the electrodes. This device has a simple structure and is particularly easy to make into an array structure, so many studies have used this structure when developing an array-type tactile sensor [54,55,56,57,58]. Even though there are various tactile sensors with several principles, they have several limitations in their applications in GR technology since the data acquisition system is unable to fully analyze the motion-detection data and capacitance data are hard to handle in constructing a measurement system compared to the current or voltage system. Furthermore, sensor arrays using these three types of transduction mechanism are controlled electrically by addressing the passive matrix, which cannot prevent the low contrast ratio and the crosstalk effect. Therefore, there is a need to identify solutions to these issues and provide direction for the future development of the technology.

### 2.3. Field Effect Transistor (FET)-Based Tactile Sensor Arrays

Gesture recognition and motion sensing require a specific process, such as acquiring a large amount of data in an array rather than using a single sensor, and this suggests the importance of fabricating a tactile sensor as an array [59,60,61]. At this time, in order to manufacture a device in the form of an active matrix, an FET array must be added to each sensor pixel. The FET is a transistor that controls the magnitude of current using an electric field applied inside the device. Therefore, by placing the FET in each pixel and connecting electrodes with terminals of the FET to measure the current flowing, independent information can be obtained from each pixel. Since the integration of an FET as a component of a tactile sensor array was reported by the Someya et al. (2014) [62], the role of FETs in tactile sensor arrays has begun to grow.

An FET-type pressure sensor can exploit the advantages of active-matrix sensor arrays that allow high-array uniformity, high spatial contrast, and facile integration with electrical circuitry [63]. Figure 3 shows typical tactile sensor arrays utilizing an FET. First, the FET can be added on the source or drain electrode, so that the current modulation has a direct effect on the drain current. Second, FET materials can be reconstructed using the piezoelectric or triboelectric materials as channel materials within the FET. Third, the gate dielectric layer of the FET can be modified as an elastomer, such as polydimethylsiloxane (PDMS, Dow Corning) or Ecoflex 0030. By integrating FETs with the pressure-sensor array, large-scale fabrication is made much easier and the consumption of power is reduced due to the extremely low off-current compared to the previously-mentioned pressure sensors. Further, the intrinsic implementation of FETs enables the device to operate as an active-matrix array; therefore, the specific characteristics of an active matrix, such as low crosstalk and high contrast ratio, are accompanied by the integration. As a result, FET-type pressure sensors make it easy to realize future sensor systems in terms of the GR.

## 3. Pressure-Sensitive Transistor Arrays

Pressure-sensitive transistor arrays were first theorized using a transistor as an electronic on–off switch for tactile sensing applications. The presence of an isolated, pressure-sensitive component that stimulates and controls the drain current of the transistor array can be classified as an extrinsic, pressure-sensitive device. Someya et al. (2004) demonstrated the nature of this application using a pressure-sensitive component as an intermediate resistance variable between the source and drain electrodes [62]. The change in the value of the drain current of the transistor arrays with respect to the resistance variations in the pressure-sensitive component when strain was applied has provided a 2D visualization of pressure-sensing detection. The integration of pressure-sensitive rubber (PSR) and organic field effect transistors have been reported to be first modifications of such tactile sensing applications. In recent work, there have been attempts to evaluate the idea of a transistor array-integrated sensor device. The concept of an intrinsic, pressure-sensitive device implements direct dynamic change in the electrical output of field effect transistors upon onset of physical strain, thereby increasing the on–off ratio and sensitivity of such sensor devices. By removing the need for the integration of pressure-sensitive components, intrinsic pressure-sensitive devices are less complex and provide superior mechanoelectrical output readings. This novel pressure-sensitive FET device has shown enhanced detection capabilities, detecting even up to cellular motion [15]. These sensor devices have achieved high signal-to-noise ratios in active-matrix pressure detection while maintaining their excellent mechanical properties. Table 1 presents the representative studies on the active-matrix transistor arrays for tactile pressure sensing.

### 3.1. FET-Based Tactile Sensor Arrays with Pressure-Sensing Components

There are two strategies involved in the application of extrinsic pressure-sensitive transistor devices in active-matrix motion sensing. First, the structural design and layout implementation of pressure-sensing components are specified in a way that yield mechanoelectrical readings that correspond to an applied force. These components are connected to data scan lines that transfer their mechanoelectrical outputs into transistor arrays. These transistor arrays enable the determination of the pressure-sensing specifications and sensitivities of these pressure-sensing devices. Another experimental strategy uses novel materials that are susceptible and receptive to physical stimuli. These novel materials intrinsically enable electrical signals when paired with a transistor array, which enhances the capability to sense pressure. However, these transistor devices are not affected dynamically by mechanical stimuli; instead, the electrical signal from the drain current is enhanced when these pressure-sensing components are deformed. Other work on pressure-sensing devices has used piezoelectric [68,69,70], capacitive [71,72], triboelectric [73,74], piezoelectric [75,76], and optical methods [77,78] as the mechanoelectrical mechanism to detect physical stimuli. However, the use of such materials in pressure-sensing devices has shown overall mechanical limitations. These materials also have shown difficulty in producing large-scale active-matrix application for tactile pressure sensing compared to transistor-based sensor devices. Transistor-based pressure sensors have been proven to have lower power consumption enabled by high on–off ratio, low signal cross-talk, and dynamic mechanoelectrical physical detection, as well as enhanced sensitivities of the sensor devices. When implemented onto thin polymer films, these devices showed a promising degree of flexibility and transparency.

#### 3.1.1. Transistor with Structure-Modified, Pressure-Resistive Component

Due to their intrinsic high on–off ratios and superior mechanoelectrical properties, transistors are used in tactile pressure sensors as electrical signal enhancers to pressure-sensing components. With the increasing interest in flexible sensor electronics, an array of transistors is placed on thin polymer films to fabricate ideal flexible sensor devices. Structure-modified extrinsic pressure sensors are designed in a way that the array of transistors is isolated from the pressure-sensing components. The array of transistors works solely as an electrical signal enhancer for the mechanoelectrical outputs of pressure-sensitive components. The change in the value of the drain current with the onset of external pressure via these components is the basic principle that governs the detection of the active-matrix pressure of these pressure sensors.

One work that demonstrated the use of a structure-modified, pressure-resistive component was the implementation of a transistor matrix array with a resistive tactile sensor (Figure 4a). Kaltenbrunner et al. (2013) demonstrated a 12 × 12 active-matrix tactile sensor array by integrating the transistor array with the physically receptive components [63]. This tactile pressure sensor has shown promising mechanical properties due to its ultra-thin dimension and deformation durability. The authors reported a high on–off ratio (~10^7^), indicating low power consumption, which is similar to most transistor-based sensor devices. This device can detect pressures are low as 1 kPa when using the pressure-sensitive, rubber-layer lamination. However, this device has limitations in detecting objects with non-conducting surfaces. Hence, pressure-sensitive rubber is laminated onto the tactile sensing array to detect non-conducting objects. Another work by Sun et al. (2014) used a 2D graphene sheet over a serpentine source channel as a structural modification to an array of tactile transistors [59]. The use of graphene enabled high transparency (80%) while maintaining excellent mechanical properties, such as a sensitivity of 0.12 kPa^−1^ and durability exceeding 2500 cycles. The 4 × 4 matrix showed a minimal pressure detection down to 5 kPa and a maximum of 40 kPa as the contact region between the source channel and the graphene sheet increased gradually after 5 kPa. However, the use of the ion-gel gate dielectric in this device limited its high range pressure detection. Similar to most transistor-based tactile sensors, this device had low power consumption, which was enabled by its high on–off ratio. Dagdeviren et al. (2014) developed a similar structure-modified transistor array sensor using piezoelectric material (Figure 4b) [79]. The device uses an 8 × 8 array of lead zirconate titanate (PZT) as the physically receptive component to a transistor amplifier for drain current output. This array is connected to the gate electrode of an adjacent FET, which enables the capability to sense pressure. The piezoelectric responses from the PZT with the onset of pressure control the value of the gate voltage of the connected transistor, ultimately giving a range of drain current values respective to their applied pressures. Although this device had a superior sensitivity of 0.005 Pa^−1^ and a response time of 0.1 ms, it was unable to detect tactile pressure when the PZT array was connected to a single transistor. Hence, the operational model of this device is not feasible for the detection of tactile pressure in a large area. The work by Chen et al. (2017) demonstrated a similar mechanism, but they used silver nanowires to alter the value of the drain current of the connected transistor [80]. The mechanical deformation of PDMS containing silver nanowires (AgNWs) induces a resistance change in the gate channel. This ultimately affects the drain current value in the transistor, thereby enabling the detection of mechanoelectrical pressure. The mechanical property of this pressure device seems relatively inferior to the previously mentioned devices, e.g., it exhibited a sensitivity of 1090 kPa^−1^, a response time of 10 ms, and a low sensing range of 0–10 kPa. However, when selenophene–diketopyrrolopyrrole donor–acceptor conjugated polymer (PSeDPP) was used as the intermediate layer in the transistor, nonvolatile memory storing capability was enabled, thereby suggesting a new direction towards advanced functional applications.

#### 3.1.2. Transistor Arrays with a Material-Modified, Pressure-Sensitive Component

Another strategy engineered to create transistor-based, tactile, sensor arrays with pressure-sensing components is to incorporate novel materials that are highly susceptible to physical stimuli into the infrastructure of the transistor. Pressure-sensitive rubber and piezoelectric, ferroelectric, and carbon-based materials commonly are used to enhance or induce the drain current values of transistors in functional sensor devices. The incorporation of these materials induces mechano-electrical readings in arrays of transistors that are related to the physical stimuli applied to the array of transistors. However, devices that are fabricated by orienting the materials of the pressure-sensitive components have increased parasitic capacitance due to the metal interconnections, limited mechanical durability, and limited response speeds.

The first ideal development of an active-matrix array of transistors incorporated with pressure-sensitive components was undertaken (Figure 4c) [62]. A 32 × 32 array of organic field-effect transistors was integrated with pressure-sensitive rubber. The increases in the applied pressure and strain reduced the resistance of the PSR, thereby increasing the value of the drain current of its corresponding transistor. The correlation between physical stimuli and the output drain current enables the transistor array to function as a tactile pressure sensor. Because the sensitivity and response time of a transistor is limited by the PSR, a relatively inferior data collection time was observed in this study when the mechano-electrical mechanism was used. Despite the disappointing performance, high on–off ratios (~10^6^) have been reported for this device, as is the case for most transistor-based pressure-sensor devices. Takei et al. (2010) reported a tactile pressure-sensor device that used a pressure-sensitive-rubber integrated field-effect-transistor array (Figure 4d) [81]. This 19 × 18 matrix sensor used parallel Ge/Si NW arrays as the active-matrix backplane to achieve superior mechanical properties despite its slow response time, i.e., ~0.1 s. The conductance change in the PSR, via the shortened tunneling path between the carbon nanoparticles when a physical stimulus is applied, induces high sensitivity to the normal pressure that is applied to the tactile sensor. However, the main limitation of this device seems to be the highly sophisticated steps in its integration process. Defective pixels frequently occur during the production of devices due to the defective sensitivities of the materials used in NW printing and the lithography procedure. Like all transistor-based tactile pressure sensors, this device has low power consumption (<5 V) and superior pressure-sensing capability (2–15 kPa). Yeom et al. (2015) reported a similar device with 20 × 20 active-matrix sensor device that uses a single-wall, carbon-nanotube thin-film transistor backplane with PSR as its conductive pressure-sensitive layer (Figure 4e) [82]. This tactile sensor shows promise for application in large-area tactile pressure sensing, with its pressure range of 1 to 7.2 kPa. With its physically durable, carbon-based layer, the device also demonstrated moderate durability during bending. Its mechano-electrical sensing capability is also notable with a response time of less than 30 ms during 1000 operation cycles. However, similar to the device constructed by Takei et al.’s group, the device exhibits high yield of defective pixels, which is induced by its sophisticated production procedures. The use of organic light-emitting diodes (OLED) in a pressure sensor based on a tactile array of transistors was demonstrated by Wang et al.’s group (2013) [64]. In addition, their device incorporated PSR and OLED layers into the bottom, thin-film transistor backplane as pressure-sensitive components. These 16 × 16 active-matrix devices use light emitted by OLED, with the intensity of the light corresponding to the magnitude of the pressure that is applied. The PSR acts as a pressure contact surface, which enhances the yield of energy transfer from the applied pressure to the intensity of the light emitted by the OLED. The light emitted by the OLED is irradiated onto an indium tin oxide (ITO) electrode that is connected to the data line of the transistor. Using light as an intermediate form of data transfer allowed the device to achieve visual tactile pressure mapping and relatively low response time that was comparable to those of other PSR incorporated devices (i.e., ~1 ms). Likewise, the transistor-based pressure-sensor device enhanced the range for sensing pressure to 5–98 kPa. Although the instantaneous visual representation of tactile pressure sensing without secondary data processing is favorable in a practical platform application, the presence of the multi-layer composition and the fragile ITO layer limit its flexibility and mechanical durability. Sekitani et al.’s group (2009) bonded a conducting PSR film onto a floating-gate transistor array sheet to fabricate a non-volatile, tactile pressure sensor (Figure 4f) [84]. This 26 × 26 active-matrix device provides an independent PSR layer between the electrode sheet and the organic memory sheet to enable the application of erasable memory. The intermediate, pressure-sensitive rubber sheet acts as an element of a pressure-sensitive component to the transistor memory sheet below. The combination of the memory transistor array sheet and the pressure-sensor rubber (PSR) sheet enables the detection of tactile pressure. Since the novelty of this device was to allow the visualization of nonvolatile 2D tactile pressure, its ability to sense active pressure was somewhat limited. The response time and pressure-sensing range of devices that incorporate PSR clearly show their limitations. Graz et al. (2006) created a similar, material-dependent, pressure-sensing array, but they used ferroelectric materials as the pressure-sensitive components [85]. They reported a modulation in the value of the drain current when the polymer matrix film, which contained ferroelectric pressure-sensitive material, was connected to the gate electrode and stimulated physically. The piezoelectric property of the pressure-sensing component contributes to the mechano-electrical mechanism of the device. The transistor sensor that was fabricated showed a promising on–off ratio of ~10^7^ and a significant pressure-sensing range from 2 to 400 kPa. Despite these qualities, this device lacks flexibility, transparency, and reasonable spatial resolution. Similar to the PSR-incorporated device, this work also showed slow response speed during operation, a low signal-to-noise ratio, and low sensitivity. Subsequent work by this group, in which a ferroelectric pressure-sensing component was used, showed slight modification (Figure 4g) [83]. The pressure-sensitive components were based on both piezoelectric and pyroelectric properties, and were connected to a single transistor of the sensor device. This incorporation allowed bifunctional sensing capability, i.e., detecting pressure and temperature independently within the device. The pressure-sensitive component was composed of a ferroelectric polymer that was embedded with piezoelectric nanoparticles, so it displayed both piezoelectric and pyroelectric properties based on its polarity direction-orientation. This sensor consisting of an array of transistors had a superior pressure-sensing range, i.e., 2–22 MPa, which was comparable to the group’s previous work. However, the operational stability of all piezoelectric devices is strictly limited by external environmental conditions, and the operation response time is also strictly limited. The measurement of physical stimuli by characterizing the mechano-electrical properties of pressure-sensitive components, such as PSR, has limitations in obtaining fast response speed. By existing as an intermediate sensing layer, PSR has slowed the overall operational speed of these sensor devices. The presence of a pressure-sensitive element seemingly increases the complexity of the devices and ultimately degrades the overall mechano-electrical sensing efficiency of such sensor devices. This, in addition to the naturally slow mechanical property of PSR, slows the operational response speed of pressure detection in extrinsic pressure-sensing FET array devices. Transistor arrays that have intrinsic pressure-sensing capability seem to correct this disability. Dynamic pressure detection by changing the structures of transistor arrays has simplified the overall pressure-sensing process by eliminating the need for a separate component for sensing pressure.

### 3.2. Pressure-Sensitive Transistor Arrays

An active-matrix structure is required to optimize the pressure sensor in the form of an array [86]. This means that external pressure-sensitive components must be implemented with transistors. Devices of this type accompany the benefits of an active matrix. However, these devices can exhibit some disadvantages, such as a complicated process and low spatial resolution. Therefore, we introduced a transistor array in which the components that make up the transistor are affected by pressure [35]. Unlike the devices introduced above, these devices have gate dielectric materials or channel materials that are sensitive to pressure, and the drain current is modulated by pressure without external pressure-sensitive components. The transistor array without external components has the advantage of an active matrix and can compensate for the shortcomings that have been mentioned. Triboelectric materials and piezoelectric materials were used to make the transistor array pressure sensitive, and the electrical properties of these materials change when pressure is applied [87]. Polarization occurs inside the piezoelectric material by physical deformation. Polarization is caused by the structure of the material, which leads to the piezopotential. The phenomenon in which piezoelectric potential is generated by mechanical force, as described above, is referred to as the piezoelectric effect, and piezoelectric pressure sensors use this piezoelectric effect. The magnitude of the drain current is changed by using the piezopotential directly or by using the generated electric field. Piezoelectric sensors have high sensitivity and fast response time. When the triboelectric material is in contact, electrostatic induction occurs through contact electrification to form a triboelectric charge. A potential difference occurs between two electrodes due to the triboelectric charge. When the triboelectric charge generates a current, the pressure can be measured. The triboelectric sensor has the advantages of low cost, easy process, and simple mechanism. Having the dielectric layer deformable by pressure is another way to make a pressure-sensitive transistor array [67]. In this case, an elastic material generally is generally used as the dielectric layer. When the dielectric layer is deformed under pressure, the distance between the gate and the channel of the transistor changes. The decrease in the distance due to mechanical force causes an increase in the capacitance of the device. The width of the channel increases due to higher capacitance, which results in increased drain current. The pressure applied can be detected by measuring the change in the drain current. This type of pressure-sensitive transistor array has the advantages of fast response time and a wide detection range.

#### 3.2.1. Options for Transistor Channel Materials

In general, transistors with pressure-sensitive channels use materials with piezoelectric or triboelectric properties. Piezoelectric or triboelectric materials are used in the form of strain-gated transistors. Chen et al. (2016) fabricated strain-gated field effect transistors using MoS_2_ and ZnO (Figure 5a) [86]. The transistor with ZnO nanowire placed over the channel makes the device pressure sensitive. When piezoelectric ZnO is pressed and polarization occurs inside, an electric field is generated as presented in other work [88]. Therefore, the drain current increases as the width of the channel increases. The pressure measurement was possible up to about 6.25 MPa. Yang et al. (2016) used triboelectric properties to fabricate a pressure sensor (Figure 5b) [89]. The device is called a tribotronic transistor array (TTA). Polytetrafluoroethylene was used as a mobile friction layer to induce charge upon contact with the electrode under pressure. The sensitivity of the device is 1.029/mm, and it can detect distances up to 9 mm based on the distance between the electrode and the polytetrafluoroethylene (PTFE).

Piezotronic transistors, which have been made simpler using piezoelectric properties in the form of conventional transistors, also are being studied [65,90]. Piezotronics basically have a structure in which a semiconductor material is placed between metal electrodes. A semiconductor is piezoelectric and usually made in nanowire form. When a piezoelectric nanowire is inserted between two metal electrodes, it may contract or relax under pressure. At the interface where the metal and semiconductor are in contact, the movement of charge is limited by the barrier. When the deformation occurs, a piezopotential is generated by nonmobile ions in the nanowire. This piezopotential changes the height of the barrier at the interface. As the height of the barrier decreases, the transfer of charges at the interface becomes easier. The basic principle is that the change in the magnitude of the piezopotential due to deformation controls the transfer of charge. Piezotronic transistors differ from conventional FETs in two ways. First, the gate component is not required because the channel material controls the charge transfer in the channel. Second, charge transfer is controlled by the movement of the charge at the interface, not by the width of the channel.

Wu et al. (2013) published a study on transistor-based pressure-sensor arrays using piezotronics (Figure 5c) [66]. A conventional transistor is a three-terminal device that includes a gate. A dielectric and a gate are required to form a channel between the source and drain. In this study, two terminal devices were fabricated by removing the dielectric and the gate using piezoelectric nanowire. A ZnO piezoelectric nanowire was placed vertically between the top and bottom electrodes of the device to make contact. The response time of the device was 0.15 s, and its sensitivity was 2.1 μS/kPa. The observed sensing range was from a few kPa to 30 kPa. Liu et al. (2017) used a ZnO nanoplatelet for the channel material to compensate for the shortcomings of nanowire (Figure 5d) [91]. When the channel material is inserted into the piezotronic transistor in the form of nanowires, lateral deflection may occur due to external pressure. This is called the buckling effect, which limits the magnitude of the piezopotential generated in the nanowire. In order to prevent the buckling effect, transistors were fabricated by making ZnO into hexagonal nanoplatelets instead of nanowires. The response time of this device is very short, i.e., less than 5 ms, and the sensitivity ranges from 60.97 to 78.23 meV/MPa. It also has a detection range from a few kPa to 3.64 MPa. Wang et al. (2017) showed a device with significantly increased sensitivity (Figure 5e) [88]. ZnO was used as the piezoelectric material, and it was made into a twin nanoplatelet to increase the sensitivity by a factor of 50. This structure lowers the barrier at both interfaces of the top and bottom electrodes. Thus, even small pressure changes can increase charge transfer significantly. The sensitivity ranged from 1448 to 1677 meV/MPa, which is a very high value. The response time was less than 5 ms, and the detection range was from 24.84 to 152.88 kPa. Conventional piezotronic pressure sensors consist of a top electrode, a bottom electrode, and a piezoelectric material between the two electrodes (Figure 5f). Liu et al. (2018) studied piezotronic transistors in which the electrodes were placed on the bottom, unlike the elements mentioned above [92]. The top electrode was removed to change from a three-layer structure to a two-layer structure. This makes the fabrication process simpler and can increase reliability in large-scale production. In addition, it shows a pressure-sensitivity range from 84.2 to 104.4 meV/MPa, which is 1.1 to 1.7 times higher than previous piezoelectric pressure sensors.

#### 3.2.2. Options for the Dielectric Materials of the Transistor Gate

Field effect transistors (FETs) operate based on the principle of flowing electrons or holes in the channel by the electric field generated by the applied gate voltage. Dielectric polarization occurs in the gate dielectric by the electric field due to the gate voltage. It forms an electric field using metal and a semiconductor as a capacitor, and the generated capacitance is expressed as follows:$$C_{i}=\frac{\epsilon_{0}\varepsilon_{r}}{d},d=d_{0}\times \left(1-\varepsilon \right)=d_{0}\times \left(1-\frac{pressure\left(P\right)}{modulus\left(E\right)}\right)$$
where capacitance is represented by permittivity $\left(\varepsilon_{0}\right)$, relative permittivity (ratio of medium permittivity and permittivity, $\varepsilon =\varepsilon_{0}\varepsilon_{r}$), and thickness of gate dielectric layer, d, when an external force is applied. Thickness, d, can be expressed by multiplying thickness, $d_{0}$, which means no force is applied to the dielectric and $\left(1-\frac{pressure\left(P\right)}{modulus\left(E\right)}\right)$ [67]. In the FET-based tactile sensor, the thickness of the gate dielectric layer decreases when pressure is applied to the gate dielectric, and, accordingly, the dielectric capacitance increases. When the gate dielectric capacitance has a high sensitivity to pressure, sensing is possible over a wider range of pressures. In addition, it could have smaller output noise and reduce errors due to its delicate acceptance of mechanical stimuli. For these reasons, various channel materials and elastomeric dielectric materials have been studied for the development of the higher pressure-sensitive gate dielectric capacitances. The characteristics of the dielectric and channel materials and the results of the modification of the dielectric layer are summarized in Figure 6 [16,93,94,95,96] and Table 2 [55,93,95,96,97,98,99,100,101,102,103,104], respectively. Sujie Chen and co-workers proposed a method of one-step processing for a large area microstructure elastomer film that had highly-integrated microfeatures of the air gap (Figure 6a). The porous PDMS was used as a microstructure elastomer film. Due to the air gap inside the PDMS, this sensor has a rapid response time at the ultra-low pressure of 1 Pa, and remains sensitive to high pressures over a wide pressure range up to 250 kPa. It also has superior durability under heavy loads, i.e., loads larger than 1 MPa [94]. D. Kwon et al. (2016) reported a pressure sensor in which they used an ecoflex-porous elastromeric material as the dielectric layer (Figure 6b). Due to the presence of micropores, this device was able to detect pressures over a wide range, i.e., from 0.1 Pa to 130 kPa. In addition, it retained its outstanding electrical properties, such as a high sensitivity of 0.601 ${\mathrm{kPa}}^{-1}$ in a low-pressure region (<5 kPa) and a high resolution of 0.17 µm [95]. Z. Bao’s group (2010) used PDMS as an elastomeric component for the first time in dielectric spacing. A low mechanical modulus is required for high mechanical sensitivity, and PDMS has a low modulus of ~2 MPa, which helps to improve the speed of the sensor’s response. The use of elastomeric materials, including air gap and the pyramid-shaped PDMS, enhances the deformation of the dielectric so that it has better sensitivity to pressure in the low-pressure region [55]. Viry et al. (2014) demonstrated a three-axial pressure sensor in which an air gap of ~150 μm was formed naturally between copper/tin-coated electrodes. This is possible due to the adhesion properties of fluorosilicone. The air gap acts as a second dielectric layer in the sensor and has high sensitivity at very low pressure (ca. 0–2 kPa) [97]. Our group proposed a simple method of forming a graphene FET pressure sensor with an air dielectric layer. We developed an origami substrate structure in which two plastic panels, one of which was the source/drain panel with the other being the gate panel, were connected to a one-foldable elastic joint, as shown in Figure 6c. By folding two plastic panels, we fabricated the pressure-sensitive graphene FET with an air dielectric layer without any damage. In addition, 50 × 50 integrated array FETs could be used for mapping the 2D pressure distribution over a wide range of pressures [67]. Z. Bao et al. (2013) reported flexible polymer transistors utilizing microstructural PDMS as a dielectric layer (Figure 6d). This pressure-sensitive, organic, thin-film transistor (OTFT) had outstanding sensitivity of 8.4 ${\mathrm{kPa}}^{-1}$, a rapid response time, high stability, and low power consumption. Due to the excellent mechanical properties of the PDMS dielectric layer, their transistors are suitable for application in electronic skin (e-skin) and health monitoring systems [96]. In addition, our group reported the fabrication of human-interactive displays (Figure 6e). Active-matrix arrays of pressure-sensitive Si transistors with air dielectric layers were fully integrated with the pixels of organic light-emitting diodes (OLEDs). Due to the air dielectric layer, pressure-sensitive transistors have high transconductance and negligible hysteresis, so the transistor can detect pressure with a relatively fast response time [16]. Zhang et al. (2015) demonstrated flexible, suspended-gate, organic, thin-film transistors (SGOTFTs), as shown in Figure 6f. The suspended gate of SGOTFTs was able to detect a wider range of pressures, and had a high sensitivity of 192 ${\mathrm{kPa}}^{-1}$ in a low-range detection pressure of <0.5 kPa. In addition, it had a low power consumption of <100 nW at an operating voltage of 6 V that could effectively detect the human pulse and the pressure of sound. In addition, one of the outstanding properties of SGOTFTs is their fast response time of 10 ms, which makes them suitable for real-time measurement devices [95].

The roughness of the oxide-channel interface of the transistor induced scattering, such as carrier trapping, which degraded the electrical properties of the transistor. To solve this problem, we used an air spacing in the dielectric layer, which allowed the gate dielectric to be formed uniformly, and this helped in avoiding the degradation of the electrical properties of the transistors. In addition, when using the air dielectric, a fast response time was possible due to the negligible hysteresis. Therefore, the elastomeric, dielectric FET-based transistor with an air gap has outstanding properties, such as fast response time, high mobility, and high sensitivity under a wide range of applied pressures, so it could be used effectively in various fields, including robotics, e-skins, and biomachines.

### 3.3. Multimodal, Multifunctional Tactile Sensor Arrays

Development of multimodal, multifunctional FET-based tactile sensors imitating multifunctional properties of human skin that can simultaneously detect various stimuli, or providing the ability of human-readable output through sensing, remain important challenges. Recently, various properties of multimodal, multifunctional tactile sensors have been achieved. Detecting normal and shear forces separately, measuring external forces simultaneously in real-time, or visualizing sensing output with emitting lights are representative examples. In this section, various studies about multimodal, multifunctional tactile sensor arrays will be presented.

Recently, we fabricated tactile sensor arrays consisting of a mechanoluminescent layer and air-dielectric ${\mathrm{MoS}}_{2}$ transistors. Cu-doped ZnS (ZnS:Cu) phosphor particles were used as mechanoluminescent (ML) components. When an external force is applied to the ML, light is emitted to the channel, and absorption of light causes a photocurrent to enable high pressure sensitivity in a wide detectable range of pressure as shown in Figure 7a [15]. The Ali Javey group demonstrated multimodal tactile sensors with active-matrix OLEDs for the first time. Outputs of applied pressures to tactile sensors are converted to human-readable information by active-matrix OLEDs. Red, green, and blue pixels comprise the OLEDs and emit light proportional to the intensity of applied pressure (Figure 7b). Pressure-sensitive rubber (PSR) is laminated on the top of the OLEDs, and its conductivity increases by applied forces [64]. In addition, Someya and co-workers made pressure- and thermal-sensitive sensors based on an organic semiconductor. By forming a net-shaped structure of a plastic film with organic transistor-based electronic circuits, conformability is possible [98]. Moreover, the Caofeng Pan group developed a polyimide-based 3D integration structure that allows simultaneous monitoring of seven physical stimuli and demonstrated stretchable multifunctional integrated sensor arrays in which each sensor can be driven independently without crosstalk (Figure 7c) [99]. E. Hwang and co-workers made flexible tactile sensors using polyimide and PDMS. The thin metal strain gauge incorporated in the polymer is used to measure normal and shear force. Each unit tactile sensor in 8 × 8 tactile sensor arrays evaluates deformations for normal and shear forces. In addition, the sensors have been proposed to measure ground reaction force because of their rigid structures [100]. D. Choi and co-workers used a pyramid-plug structure to detect normal force, shear force, and torsion. By utilizing pyramid-shaped engraved electrodes and ionic gel as neural mechanoreceptors, each mechanical force has its own deformation mechanism, as shown in Figure 7d [101]. Similarly, J. Park et al. developed stress-direction-sensitive, stretchable e-skin inspired from interlocked microstructures found in epidermal-dermal ridges in human skin. Microdome arrays employed for e-skin have a highly sensitive detection capability for various mechanical stimuli, such as normal, shear, stretching, bending, and twisting forces (Figure 7e) [102]. For commercialization of multimodal, multifunctional tactile sensors, the Rogers group proposed the fabrication of epidermal electronics using multimodal, multifunctional tactile sensors for the first time. Unlike basic wafer-based technology, conformal contact is possible using only Van der Waals force on curved skin. By attaching their devices to the back side of a commercial temporary transfer tattoo, they achieved multifunctional invisible epidermal electronic skins as shown in Figure 7f [103].

In short, enhancing the properties of multimodal, multifunctional tactile sensor arrays, such as by monitoring diverse stimuli at the same time, or visualizing the outputs of tactile sensing data, provides potential applications in human-health monitoring and biomachine interface fields, as well as robotics and e-skin.

## 4. Future Applications for the Tactile Sensor Arrays Based on FETs

### 4.1. Robotics

Robotics uses an artificial machine to execute functions previously performed by humans, and can be useful in many academic and industrial fields [104,105]. Currently, robotics is mostly used in manufacturing in the industrial field. However, robotics can be applied in almost any field. Robotics enables access to dangerous and inaccessible locations in various fields such as agriculture, biotechnology, medical, electronics, and engineering. Robotics can replace humans actions, with better performance than humans in many cases. It can also perform tasks that might be limiting for humans. Recently, soft robotics using flexible materials has been studied extensively [106,107,108,109]. Soft robotics, which have flexibility, stretchability, and transparency, can have various applications, and can be used, in particular, to create a human-interactive device.

Human skin is one of the largest sensor networks that can detect various stimuli, such as pressure, temperature, strain, humidity, and pain. It also sends signals to the brain through synapses. E-skin is an electronic device that can mimic and reproduce these various characteristics of the human skin in materials, functions, and structures [110,111,112]. E-skin can be applied to robotics, and is highly useful for medical diagnostics when attached to the skin. It can also be applied to replacement prosthetic devices instead of injured skin and as a monitoring device that monitors physical condition in real time.

Human skin has mechanoreceptors that respond to mechanical stimuli [113,114]. In a mechanoreceptor, when a mechanical stimulus is received from the outside, mechanotransduction occurs that makes the stimulus into a current of ions through the ion channel. The flow of ions generated transmits information of the stimulation to the brain through synapses. A function to be noted in mechanoreceptors is that they can convert external stimuli into electrical signals. In order to transmit the external stimulus and use the detected information, it must be converted into electrical signals. Physical changes, such as the deformation of materials and the shrinkages of thickness or length that occur under pressure, are not readily available information in digital devices. If the tactile sensor array transmits information through a change in electrical information due to a change in pressure, it can be monitored, e.g., the brain receives signals from the body through the delivery of ions, and a signal is received to diagnose the meaning. In addition, such signals can be processed into useful information that can be used more easily.

Mechanical properties are important to use e-skin effectively. The component in which e-skin is used will be attached to actual human skin or robotics that can move like a human body. It must be able to bend or fold like real skin and stretch without damage. Therefore, e-skin requires both flexibility and stretchability. In order to adhere to the skin, the adhesion properties of the device must also be very good [115]. To be used in people, it must be able to withstand the ingress of moisture [116].

Park et al. (2019) fabricated ultrathin, tactile sensors based on MoS_2_ (Figure 8a) [117]. In this paper, MoS_2_ TFTs were deposited on Al_2_O_3_ (50 nm)/SU-8 (500 nm)/polymethyl methacrylate (PMMA) (sacrificial layer) and fabricated on a SiO_2_/Si substrate. Then, the PMMA was removed and the fabrication of the sensor was completed on a very thin, i.e., 2 mm thick, PDMS with a Young’s modulus similar to that of human skin. The tactile sensor was attached directly to the palm, and experiments were conducted to sense pressure for various inputs. Another study used a porous structure as a pressure-sensitive layer. Lou et al. (2017) composed a device by putting a layer of polyaniline hollow nanosphere composite films (PANI-HNSCF) between two electrodes (Figure 8b) [118]. It had a size of 10 × 10 pixels and experiments were conducted in which the device was attached to various parts. It was confirmed that it could be twisted without damage. In addition, it was attached to the back of the hand to measure pressure, and the applied pressure could be mapped. Boutry et al. (2018) fabricated an e-skin that mimics the microstructure of the skin and simultaneously can detect normal and shear forces (Figure 8c) [119]. Carbon nanotubes (CNTs) were fabricated as an array and covered with polyurethane (PU) to form a spherical microstructure of PU that contained CNTs. A sensor manufactured in this way can detect two forces through changes in resistance and capacitance between electrodes. After fabrication, an e-skin capable of pressure sensing was applied to the arm of robot. The pressure sensor was attached to the fingertip of the robot’s hand, providing feedback on the pressing action. Figure 8c shows that the robot’s hand does not break the raspberry due to feedback. However, there is no feedback in the picture below, which confirmed that the raspberry was broken by pressure. As such, it can be used as a role for feedback on an action. Research has also been conducted on the recognition of braille by applying a pressure sensor at the fingertip (Figure 8d). You et al. (2018) fabricated a tactile sensor array using microparticles [120]. This device, which was assembled by inserting microparticles between two electrodes, has high sensitivity even at very low pressures. Because it can be operated even at low pressure and has high sensitivity, it is suitable for use as a sensor applied to the tip of the finger for research purposes. The device was made of 10 × 10 pixels and was used to recognize a braille sign stating “restroom for handicapped males”. The arranged braille was detected accurately by a 10 × 10 pixel sensor. Sundaram et al. (2019) made a scalable tactile glove (STAG) that covered the entire palm with a pressure-sensor array (Figure 8e) [121]. They analyzed the behavior of interacting with 26 objects. The array of 548 sensors was assembled on a knitted glove, and a large-scale tactile dataset with 135,000 frames was recorded. It was confirmed how the pressure acts on each part of the palm depending on the objects. This work proved the worth of tactile sensors, as well as emphasizing the potential for robotics.

### 4.2. Artificial Intelligence (AI)

For the commercialization of tactile sensors, FET-based tactile sensors with outstanding electrical and mechanical properties must be researched and developed, and they must be able to accept and recognize various stimuli of the features of unknown objects through machine learning. Tactile sensors with machine learning can judge how to deal with various unknown objects. This helps the development of future industrial applications that can create self-made solutions to the output after taking measurements, rather than just measuring the stimuli. Spiers et al. (2014) proved that the recognition of an object and the extraction of its features are possible in the sensing process of tactile sensors with machine learning through the training of random forest classifiers. A robot hand with machine learning has high-accuracy data classification and an excellent ability to determine the characteristics of objects [122]. Liu (2016) developed the ability to represent and classify tactile data using the kernel sparse coding method. The use of existing sparse coding has led tactile data to deal with the interference problem when all of the robot’s fingers touch an object at the same time. They solved this problem by developing the joint kernel sparse coding model [123]. To analyze the motion of an unknown object, Luo analyzed the features of the object using a new tactile-SIFT (scale-invariant feature transform) descriptor [124]. Mohsen et al. (2017) developed a multi-functional tactile sensor that was capable of sensing multiple stimuli and made it possible to recognize the features of unknown objects through machine learning [125].

To date, various studies of AI tactile sensors capable of the recognition and acquisition of the motion of an object have been published. In addition, after obtaining various data (e.g., shape, color, elasticity, or roughness), research is being conducted actively to develop an application of a human–machine interactive FET-based tactile sensor capable of simultaneously classifying and making decisions about the data through machine learning.

### 4.3. Human-Machine Interface (HMI)

Multimodal and multifunctional tactile sensors developed to recognize human processes and the results of sensing data measured in real time, not just simply measuring human’s motions or external forces, have attracted significant attention recently [126,127]. Optical tactile sensors enable interactive viewing through eyes, which is the most basic method for recognizing data. Thus, the importance of HMI tactile sensors is increasing due to research on devices that can visualize outputs. In this section, we introduce optical tactile sensors as representative examples of HMI tactile sensors.

Optical tactile sensors can reduce stray capacitance (e.g., unwanted capacitance that occurs due to sensor proximity) and thermal noises. In addition, they can reduce the complexity of wiring and crosstalk, thereby enabling high spatial resolution [128]. X. Wang et al. (2015) demonstrated a pressure-sensor matrix capable of recording the pressure, force, and speed of signing applied to each pixel in personalized signing. Pressed mechano-luminescent materials emitted light and were visualized by a pressure-sensor matrix. A piezophotonic mechanism was used to operate this device [78]. E. H. Kim et al. (2019) demonstrated an interactive skin display with an epidermal stimuli electrode. It was based on a two-layer structure with light-emitting inorganic phosphors in a structure with gaps in the two electrodes. The interactive skin display with an epidermal stimuli electrode visualized the motion sensing of the tactile sensor while stimulation was applied by external forces [129]. Our group provided a platform for wireless pressure sensing with a built-in battery and instant visualization. Minimization was achieved by simply using a built-in battery on an FET-based pressure sensor that consisted of a Si channel and an air dielectric. In addition, it made it possible to visualize the real-time output with a smartphone by processing with a wireless device [126].

Devices that have been optimized to operate an interactive display for diverse external stimuli are proposed as potential applications in various fields relative to biomotion, such as health care and biomachines. Current studies can help develop HMI tactile sensors with characteristics of olfactory or auditory capabilities in addition to visualization in the next generation of these devices.

### 4.4. Healthcare Biosystems

The demand for emerging flexible electronic technologies, such as wearable healthcare devices biomedical prosthesis, robotics, and electronic skin (e-skin), has resulted in profound research interests in developing tactile pressure sensors [130,131]. One of the main reasons for GR is to create a people-friendly environment. For this purpose, motion detection for healthcare systems and biosystems has also become an important research goal, and there are several challenges that must be overcome to produce such a device for biological objects. For example, pressure sensing and monitoring of biological systems, such as cardiac and vascular tissues, joints, cartilage, and spinal cord discs, have clinical importance that requires high-performing sensors in terms of sensitivity and accuracy, wide sensing range, enhanced spatial resolution, and low power consumption. To meet these requirements, many different types of pressure sensors, such as resistive, capacitive, and piezoelectric pressure sensors, have been developed in recent years using various nanomaterials with different structural configurations [132,133,134]. Despite these advancements, utilization of flexible devices is still limited to specialized applications that require specific sensing ranges, sensitivities, and spatial resolutions, and these devices are inadequate for generalized application in bio-systems.

Kim et al. (2017) reported a wearable smart contact lens with highly transparent and stretchable sensors that continuously and wirelessly monitor glucose and intraocular pressure, which are the risk factors associated with diabetes and glaucoma, respectively [135]. In this work, the authors wirelessly demonstrated real-time, in vivo glucose detection on the eye of a rabbit and in vitro monitoring of the intraocular pressure of a bovine eyeball. This device is the completed form of an integrated system of a pressure sensor and a biochemical sensor, suggesting a substantial platform for future biomedical applications of tactile pressure sensors [136]. Jang et al. (2020) presented the formation of active-matrix, pressure-sensitive, MoS_2_ FET arrays with air dielectrics and mechanoluminescence materials for sensing extensive ranges of tactile pressure [15]. In this work, a single FET can operate solely as a pressure sensor, and, therefore, the active-matrix formation using these FETs can be advantageous for high spatial resolutions with miniaturized integrations. In addition, these approaches could enlarge the detectable range of pressure from 70 Pa to 5 MPa by successfully monitoring the pressure distributions of single cell motions and heel footsteps in real time. In particular, the outstanding spatial and temporal resolutions of this active-matrix sensor array enabled single- and/or sub-cellular pressure mapping. This sensor array shows potential for being beneficial for pressure sensing and monitoring multiple events in biology for future applications of prostheses, robotics, and various biomedical electronics, from muscles, joints, and cartilage working under high pressure to cardiac and vascular systems under mild and small pressures. Based on the results of these studies, it is expected that research that confirms the correlation by measuring and analyzing the results of mechanical pressure rather than biological potential with a tactile sensor will be actively conducted.

## 5. Prospects

As previously discussed, FET-based tactile pressure sensors have advantageous features, such as high pixel resolution, wide range of detectable pressure, fast response time, and reliable performance, and are suitable for numerous applications, including GR technology. In addition, active-matrix arrays, which are inherent properties of FET arrays, can provide a tactile sensor device with low power consumption, low signal crosstalk, and a high contrast ratio. In addition, multi-functional tactile sensors exist that can convert tactile stimuli into visual and auditory signals, suggesting substantial potential for future robotics and motion-detection systems. The tactile sensor devices based on distinct functions and characteristics can be used in various fields, such as robotics, artificial intelligence, human-machine interfaces, and health monitoring systems.

Nonetheless, there are challenges that must be overcome for the recognition of gestures and the detection of motion, and some of these challenges are presented here. First, sophisticated multi-modal and multi-functional sensors are required to detect the various types of stimulus from people’s movements. Detection and analysis of a single pattern of tactile stimuli cannot provide sufficient information for a computing system to recognize various gestures. Second, excellent mechanical stability against stretching and bending is required for wearable electronics. Since robotics and HMI technology have attracted the attention of researchers in various fields, tactile sensors can be applied directly to a robot’s parts, such as the hand and arm. In this case, it is important to maintain the robot’s sensing characteristics during the various types of mechanical deformations. Third, superb biocompatibility of the materials that are used is essential so that the robot can operate in harsh environmental conditions, such as sweat and water. In addition, for e-skin applications, materials that are incompatible with biosystems do not easily interact with the human body because they can cause inflammation and irritation of the skin. Therefore, we plan to concentrate on these challenges that must be solved in order to develop advanced GR technology.

## 6. Conclusions

In conclusion, excellent research on FET-based tactile pressure sensors has been conducted, followed by accelerated development of pressure sensors, including touch interfaces. Although non-contact interface-based gesture recognition has developed rapidly, environmental restrictions inhibit the application of the systems in practice. In addition, the principles underpinning various tactile sensors have their own advantages and disadvantages, suggesting the sensor type should be appropriately customized and arranged to fulfil target applications. However, the current direction of development is toward interactions frequently performed by humans and machines (e.g., motion detection and gesture recognition), suggesting the implementation of the active matrix, which provides the advantages of low power consumption and signal crosstalk, is essential. We envision that this type of tactile pressure sensor will be beneficial for GR technology, as well as future applications, such as prostheses, robotics, artificial intelligence, and biomedical electronics, and will provide substantial opportunities to develop the next-generation motion sensing systems.

## Figures and Tables

**Figure 1 sensors-20-03624-f001:**
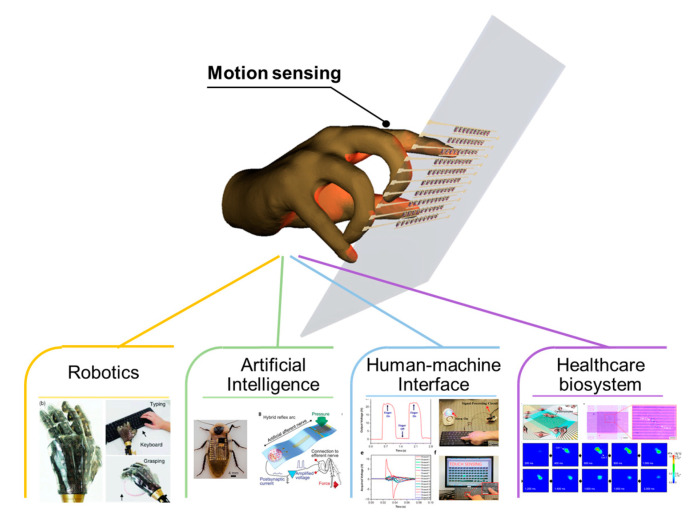
Tactile sensors for motion sensing. Reproduced with permission [24]. Copyright 2014, Springer Nature. Reproduced with permission [9]. Copyright 2018, American Association for the Advancement of Science. Reproduced with permission [10]. Copyright 2015, American Chemical Society. Reproduced with permission [25]. Copyright 2020, American Chemical Society.

**Figure 2 sensors-20-03624-f002:**
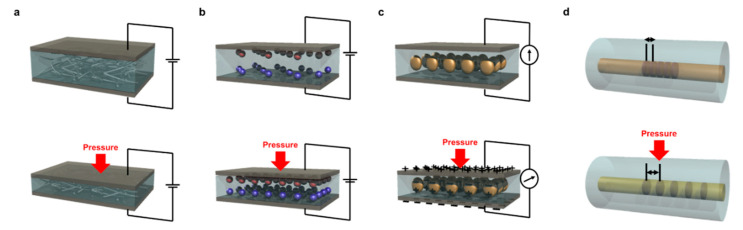
Schematic illustrations of transduction mechanisms of tactile sensors: (**a**) piezoresistivity; (**b**) capacitance; (**c**) piezoelectricity; (**d**) optics.

**Figure 3 sensors-20-03624-f003:**
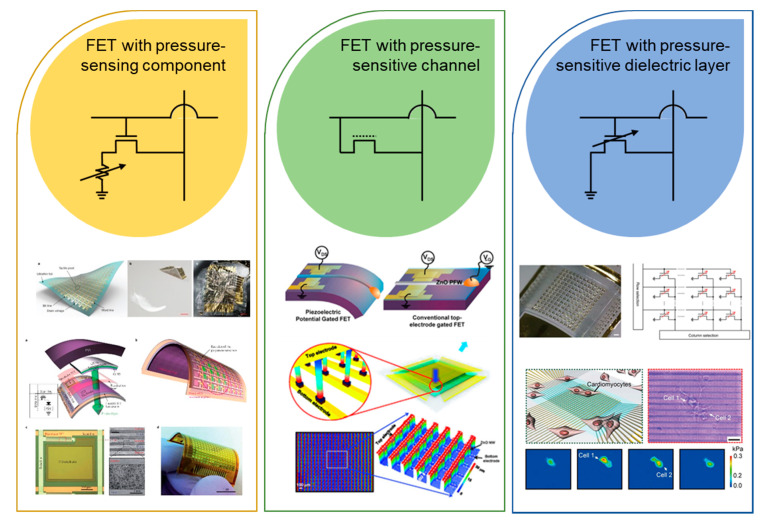
Representative tactile sensor arrays with active-matrix driving with circuits. Reproduced with permission [63]. Copyright 2013, Springer Nature. Reproduced with permission [64]. Copyright 2019, Springer Nature. Reproduced with permission [65]. Copyright 2010, Elsevier. Reproduced with permission [66]. Copyright 2013, American Association for the Advancement of Science. Reproduced with permission [67]. Copyright 2017, Springer Nature. Reproduced with permission [15]. Copyright 2020, American Chemical Society.

**Figure 4 sensors-20-03624-f004:**
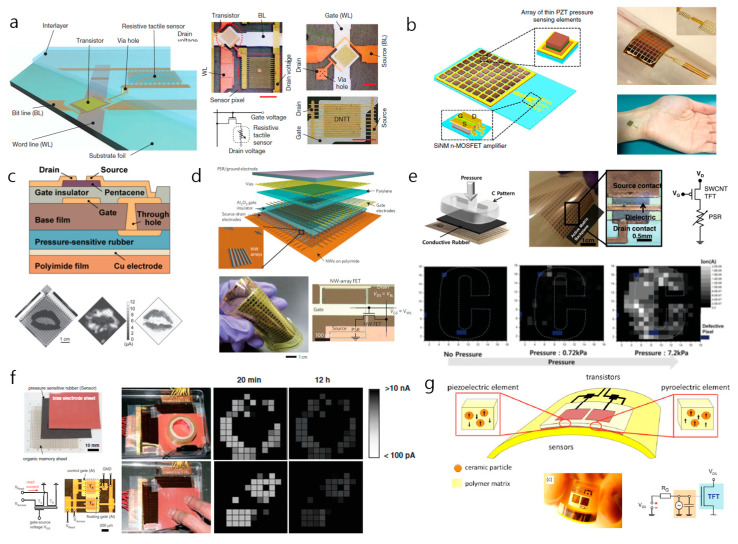
Field-effect transistor (FET)-based tactile sensor arrays with pressure-sensing components. (**a**) Active-matrix tactile pixel sensor comprising a switching transistor and a resistive touch sensor. (**b**) Thin conformal array pressure sensor made of elements of zirconate titanate (PZT) with transistor amplified pressure-sensing capability. (**c**) Tactile pressure sensor with pressure-sensitive component in its active-matrix transistor array. (**d**) Nanowire-based flexible tactile pressure sensor with pressure-sensitive rubber (PSR)-integrated active-matrix transistor array. (**e**) Tactile pressure sensor with carbon nanotube backplane integrated with active-matrix transistor array. (**f**) Nonvolatile tactile pressure sensor with active-matrix two-transistor array with intermediate pressure-sensitive rubber sheet. (**g**) Bifunctional sensor array with flexible ceramic polymer sensor laminated onto transistor backplane. (**a**) Reproduced with permission [80]. Copyright 2013, Springer Nature. (**b**) Reproduced with permission [79]. Copyright 2014, Springer Nature. (**c**) Reproduced with permission [62]. Copyright 2004, National Academy of Sciences. (**d**) Reproduced with permission [81]. Copyright 2010, Springer Nature. (**e**) Reproduced with permission [82]. Copyright 2015, John Wiley and Sons. (**f**) Reproduced with permission [84]. Copyright 2009, The American Association for the Advancement of Science. (**g**) Reproduced with permission [83]. Copyright 2009, Journal of Applied Physics.

**Figure 5 sensors-20-03624-f005:**
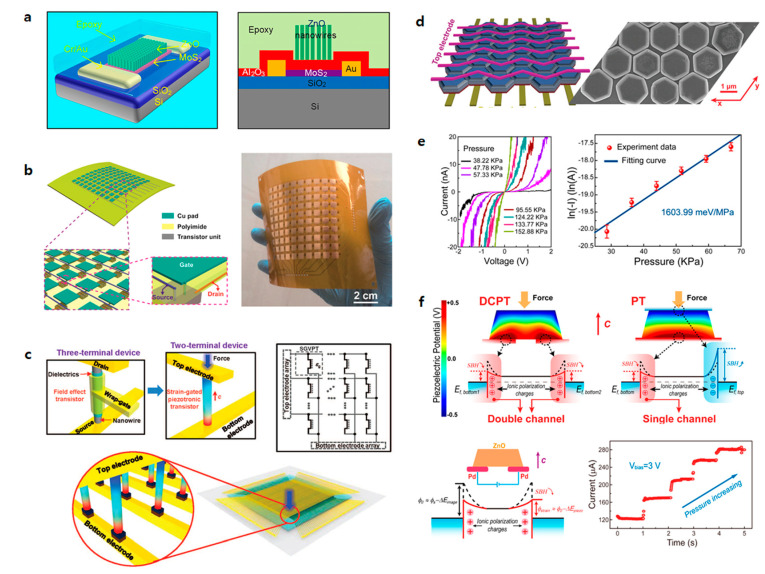
Transistor array using pressure-sensitive channel materials. (**a**) Schematic illustration of a FET based on a MoS_2_ and ZnO heterostructure (left). Schematic cartoon graphs of the architecture of the device that uses the ZnO nanowire (NW) array and MoS_2_ flake, separated by a 20-nm atomic layer deposition (ALD)-deposited Al_2_O_3_ layer. (**b**) Schematic of a 10 × 10 tribotronic transistor array (TTA) (left). Partial enlarged tilted views of the TTA configuration and pixel structure, respectively (inset). Optical photograph of a fully integrated TTA with each sensing pixel of 5 × 5 mm (right). (**c**) Comparison between three-terminal voltage-gated NW FET and two-terminal strain-gated vertical piezotronic transistor (left). Color gradient in the strained strain-gated vertical piezotronic transistor (SGVPT) represents the strain-induced piezopotential field, in which red and blue indicate positive and negative piezopotential, respectively. ZnO NWs in SGVPT grow along the c axis (red arrow). Equivalent circuit diagram of the 3D SGVPT array (right). Equivalent circuit diagram of the 3D SGVPT array (bottom). The region highlighted by black dashed lines is the unit SGVPT device, in which ε_g_ represents the mechanical strain gate signal and the vertical dotted line between the two terminals of the SGVPT denotes the modulation effect of ε_g_ on the conducting characteristics of the device. (**d**) Schematic illustration of a 2D piezotronic transistor (2DPT) array using ZnO nanoplatelet (left). Scanning electron micrograph of 2DPT array with high spatial resolution (≈12,700 dpi) (right). (**e**) The modulation of carrier transport by pressure in a piezotronic transistor based on ZnO twin nanoplatelets, which shows the characteristic of piezotronic effect (left). ln(I)−P curve demonstrates a linear relationship between ln(I) and the applied pressure, showing the extreme sensitivity and indicating the modulation effect of applied pressure on conductance (right). (**f**) Piezopotential distributions and the corresponding energy-band diagrams of double-channel piezotronic transistor (DCPT) and conventional piezotronic transistor (PT) (top). Compared to one rise and another drop in conventional PT, both Schottky barriers in DCPT decrease with increasing pressure. Energy-band diagram of DCPT without (black dashed line) and with (red line) piezotronic effect, in which ΔE_image_ and ΔE_piezo_ represent mirror force and piezopotential-induced Schottky barrier height change, respectively (left). Current increased step-by-step with increasing pressure by a step of 63.5 kPa from 0.75 to 1.00 MPa at a fixed bias (right). (**a**) Reproduced with permission [86]. Copyright 2016, American Chemical Society (**b**) Reproduced with permission [89]. Copyright 2016, American Chemical Society. (**c**) Reproduced with permission [66]. Copyright 2013, American Association for the Advancement of Science. (**d**) Reproduced with permission [91]. Copyright 2017, American Chemical Society. (**e**) Reproduced with permission [88]. Copyright 2017, American Chemical Society. (**f**) Reproduced with permission [92]. Copyright 2018, American Chemical Society.

**Figure 6 sensors-20-03624-f006:**
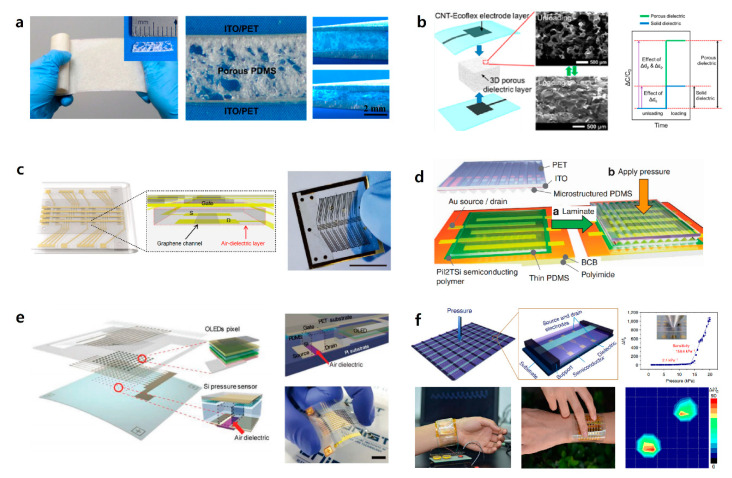
Results of dielectric layer modification. (**a**) The photo image of the fabricated large area microstructured polydimethylsiloxane (PDMS) film (the inset shows the cross-sectional photo image of the microstructured PDMS film) (left); The cross-sectional photo images of the microstructured PDMS film clipped by a tweezer without pressure (upper) and with pressure (bottom) (right). (**b**) Relative changes in the capacitances of pressure sensors using solid and porous elastomeric dielectric layers induced by identical levels of external loading. The synergy of the larger deformation by the reduced stiffness and the increased effective dielectric constant by the closure of the air gap dramatically amplifies the capacitance change. (**c**) Schematic images of pressure-sensitive graphene FETs with air-dielectric layers after the folding. The air-dielectric layer is placed between the graphene channel and the gate electrode as illustrated in the schematic image (inset) (left); Photograph of the fabricated pressure-sensitive graphene FETs; scale bar, 1 cm (right). (**d**) Schematic of the final fabrication step of our pressure-sensitive transistor. (**e**) Schematic of active-matrix pressure-sensitive display, integrating the Si tactile pressure sensors and organic light-emitting diodes (OLEDs). (**f**) Suspended gate organic thin-film transistor pressure sensors. (**a**) Reproduced with permission [94]. (**b**) Reproduced with permission [95]. Copyright 2016, ACS Applied Materials & Interfaces. (**c**) Reproduced with permission [67]. Copyright 2017, Nature Communications. (**d**) Reproduced with permission [93]. Copyright 2013, Nature Communications. (**e**) Reproduced with permission [16]. Copyright 2019, Advanced Material Technologies. (**f**) Reproduced with permission [96]. Copyright 2015, Nature Communications.

**Figure 7 sensors-20-03624-f007:**
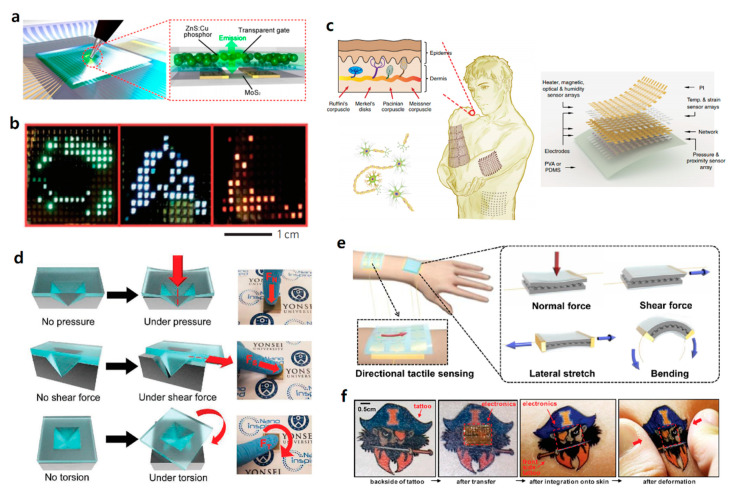
Multimodal, multifunctional tactile sensor arrays based on FETs. (**a**) Schematic layouts of the pressure-sensor array integrated with ZnS:Cu phosphor particles. (**b**) Green, blue, and red color interactive e-skins are used to spatially map and display the pressure applied with C-(left), A-(center) and L-(right) shaped PDMS slabs, respectively. (**c**) Skin-inspired highly stretchable and conformable matrix networks; a schematic illustration of stretchable and conformable matrix networks (SCMNs) conforming to the surface of a human arm and an expanded network (expansion: 200%) conforming to the surface of a human abdomen (right); the tree branch-like connections of neurons (left bottom); the sensory receptors of the glabrous skin (left top). (**d**) Schematic illustrations without stimulus and under three different mechanical stimuli for pressure, shear force, and torsion. There were possible geometric deformations of the pyramid-plug structure with mechanical loadings. (**e**) Schematic of a stress-direction-sensitive electronic skin for the detection and differentiation of various mechanical stimuli including normal, shear, stretching, bending, and twisting forces. (**f**) A commercial temporary transfer tattoo provides an alternative to polyester/polyvinyl alcohol (PVA) for the substrate; in this case, the system includes an adhesive to improve bonding to the skin. Images are of the back side of a tattoo (far left), electronics integrated onto this surface (middle left), and attached to the skin with electronics facing down in undeformed (middle right) and compressed (far right) states. (**a**) Reproduced with permission [15]. Copyright 2020, Nano Letters. (**b**) Reproduced with permission [64]. Copyright 2013, Nature Materials. (**c**) Reproduced with permission [99]. Copyright 2018, Nature Communications. (**d**) Reproduced with permission [101]. Copyright 2019, Advanced Materials Technologies. (**e**) Reproduced with permission [102]. Copyright 2014, ACS Nano. (**f**) Reproduced with permission [103]. Copyright 2011, Science.

**Figure 8 sensors-20-03624-f008:**
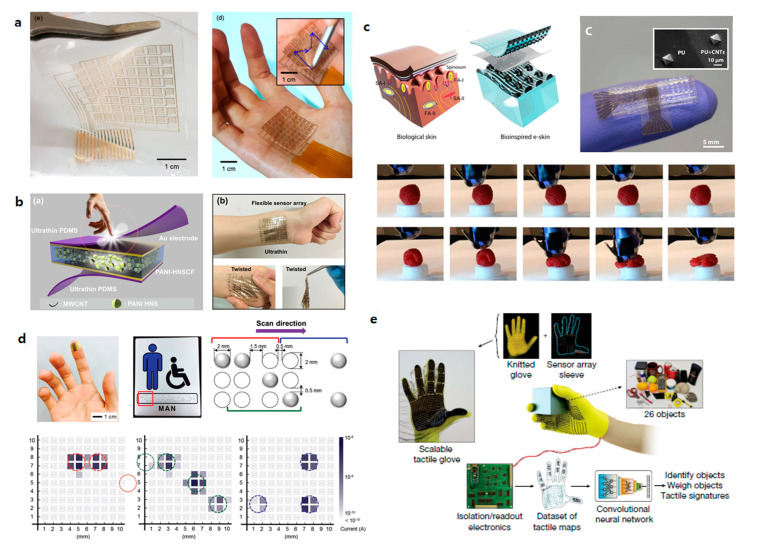
Tactile pressure sensors for e-skin and robotics. (**a**) Photograph of the large area flexible active-matrix (AM) MOS_2_ tactile sensor array (left). Photograph of the AM MoS_2_ tactile sensor on the human palm (right). (**b**) Primary design concept of flexible e-skin sensor based on polyaniline hollow nanosphere composite films (PANI-HNSCF) (left). Optical image of a wearable ultrathin e-skin sensor array on the wrist and under various mechanical deformations (right). (**c**) Cross-section of the skin of the fingertip depicting key sensory structures and soft biomimetic e-skin (left). Optical image showing carbon nanotube-polyurethane (CNT-PU) interconnects for signal recording with inductance/capacitance ratio (LCR) meter and SEM picture of the top e-skin layer with molded pyramids (right), showing CNT-PU and PU areas (inset). Tactile feedback prevented flattening of the raspberry. Without tactile feedback, the fruit was crushed (bottom). (**d**) Digital image of the flexible tactile array sensor attached to the tip of an index finger as an artificial fingertip (left). Digital image of a braille sign stating “restroom for handicapped males” (middle). Illustration denoting the dimensions of the scanned braille (right). Current profiles obtained from the artificial fingertip while scanning the braille (bottom). (**e**) The scalable tactile glove (STAG) consists of a sensor array with 548 elements covering the entire hand, attached to a custom knit glove. An electrical readout circuit is used to acquire the normal force recorded by each sensor at approximately 7.3 fps. Using this setup allows recording of a dataset of 135,187 tactile maps while interacting with 26 different objects. A deep convolutional neural network trained purely on tactile information can be used to identify or weigh objects and explore the tactile signatures of the human grasp. The glove shown at the center is a rendering. (**a**) Reproduced with permission [117]. Copyright 2019, American Chemical Society. (**b**) Reproduced with permission [118]. Copyright 2017, Elsevier. (**c**) Reproduced with permission [119]. Copyright 2018, American Association for the Advancement of Science. (**d**) Reproduced with permission [120]. Copyright 2018, John Wiley and Sons. (**e**) Reproduced with permission [121]. Copyright 2019, Springer Nature.

**Table 1 sensors-20-03624-t001:** Comparison of active-matrix transistor array device for tactile pressure sensing.

	Device Composition	Pressure-Sensing Range	Response Time	Spatial Resolution	Sensitivity	Reference
Structure-Modified	PSR	Minimal pressure down to 1 kPa		12 × 12		[63]
Structure-Modified	2D graphene sheet	5 kPa~40 kPa		4 × 4	0.12 kPa^−1^	[59]
Structure-Modified	PZT		0.1 ms	8 × 8	0.005 Pa^−1^	[79]
Structure-Modified	AgNWs	0–10 kPa	10 ms		1090 kPa^−1^	[80]
Material-Modified	PSR	2–15 kPa		32 × 32		[62]
Material-Modified	PSR		0.1 s	19 × 18	∼11.5 μS kPa^−1^	[81]
Material-Modified	Carbon-nanotube, PSR	1–7.2 kPa	<30 ms	20 × 20	~800% kPa^−1^	[82]
Material-Modified	OLED, PSR	5–98 kPa	1 ms	16 × 16	~42.7 Cd m^−2^ kPa^−1^	[64]
Material-Modified	Ferroelectric nanoparticle	2–22 MPa				[83]

**Table 2 sensors-20-03624-t002:** The tactile sensor characteristics with various gate dielectric and channel materials.

Dielectric Materials	Channel Materials	Pressure-Sensing Range	Response Time	Spatial Resolution	Sensitivity	Reference
Pressure-sensitive rubber (PSR)	50 nm-thick pentacene layer	0 to 30 kPa	~30 ms of cycle time	16 × 16 arrays 10 dpi (254 mm)		[62]
Ecoflex-porous elastomeric		0.1 Pa to ~130 kPa	Fast response time	0.17 μm Position resolution	0.601${\;\mathrm{kPa}}^{-1}$ at 5 kPa	[95]
Microstructured V-shape-groove PDMS (pyramid molding PDMS)	Rubrene single crystal	0 to ~7 kPa		~ 1 mm	(<2 kPa) 0.55${\;\mathrm{kPa}}^{-1}$ (>2 kPa) to 0.15 kPa^−1^	[55]
Microstructural PDMS	PiI2T-Si	0 to 60 kPa	< 10 ms	~ 1.7 mm	(<8 kPa) 8.2${\;\mathrm{kPa}}^{-1}$ (>30 kPa) 0.38 ${\;\mathrm{kPa}}^{-1}$	[93]
PDMS with air-gap	Graphene	250 Pa to ~3 MPa	30 ms	1 mm	(<500 kPa) to 2.05 × 10^−4^ ${\mathrm{kPa}}^{-1}$ and (>500 kPa) to 9.43 × 10^−6^ ${\mathrm{kPa}}^{-1}$	[67]
PDMS + G-PDMS (G-PDMS; synthesis of glycerol and PDMS at 1:10)	$\mathrm{Mo}S_{2}$	200 Pa to 5 MPa	25 ms	500 μm	(<500 kPa) 1.8 ${\mathrm{MPa}}^{-1}$ (>500 kPa) 0.018 ${\mathrm{MPa}}^{-1}$	[15]

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
