# Peer review of "Motion Detection Using Tactile Sensors Based on Pressure-Sensitive Transistor Arrays"

_sensors, 2020, doi:10.3390/s20133624_

Round 1

Reviewer 1 Report

Tactile sensor arrays based on transistors with an active-matrix design are more promising than other pressure sensors such as static array or passive arrays in addressing pressure distribution in real world applications because of large-area capability and low signal crosstalk. Although the high importance of tactile-sensing transistors, I was astonished that there were only a few review papers on this topic. This review on pressure-sensitive transistor arrays is in time and necessary, and will be attractive to broad readership in flexible electronics.

Still, I have a few concerns before I recommend accepting it for publishing:

  1. In section 2.2, the authors may distinguish between piezoresistive sensor (based on silicon MEMS) and resistive pressure sensors. Also, the authors may use “flexible tactile sensor” in title and the main texts.
  2. Also, there are other types of flexible pressure sensors not included (in Figure 2), for example soft optical sensor, optical fibre sensor, and triboelectric pressure sensors.
  3. In section 2.3, the basic operation principle of FET or TFT, and why transistors were the best choice for constructing active-matrix pressure-sensing arrays should be better elaborated. For example, in lines 188-189, page 5, the reason why “an FET array must be added to each sensor pixel” deserves more detailed discussion.
  4. Some images are not clear, e.g. Figure 6 and 7. The words cannot be clearly recognized.

Author Response

We thank the reviewer for the thoughtful review and comments about our manuscript, and we welcome the opportunity to address and clarify the issues raised in the referee’s report. Our responses to the comments on the report are attached as PDF file below.

Reviewer 2 Report

This paper is a well prepared overview of a tactile sensors based on  pressure-sensitive transistor arrays. At the initial stage of the paper it will be nice to show a set of Partial Differential Equations that are used at the modelling stage of such devices, and some simple numerical models and simulation results illustrating the operation principle. For example corresponding to Fig. 2. - time diagrams of pressure and output electrical signals. On the other hand, the paper is in an overview form, so can be accepted. The Editor should decide about this. 

Author Response

We thank the reviewer for comments about our manuscript, and welcome the opportunity to address and clarify the issues raised in the referee report. Our responses to the points raised in the report are attached as PDF file below.
